

# A nonparametric approach toward upper bounds to transit time distribution functions

Earl Bardsley[1]

[1]School of Science, University of Waikato, Hamilton 3240, New Zealand

*Correspondence to*: Earl Bardsley (earl.bardsley@waikato.ac.nz)

**Abstract.** A nonparametric method is proposed as a possible approach to obtaining upper bounds to distribution functions of time-varying transit times for catchment environmental tracers. A discretization is employed for the tracer throughput process, with tracer input represented as a sequence of $K$ discrete pulses over a given time period. Each input pulse is associated with a different and unknown upper-bounded nonparametric discrete transit time distribution. The model transit time distribution function is therefore a $K$-component finite mixture of different and unknown discrete distribution functions, weighted by the relative magnitudes of the respective tracer pulses. Upper bounds to this distribution function can be obtained by linear programming to achieve a sequence of $K$ discrete optimised transit time distributions which yield the maximum possible value of tracer fraction less than a given age, subject to a constraint of matching the catchment tracer output time series to some specified linear measure of accuracy. The individual optimised distributions do not estimate actual transit time distributions and the optimisation procedure is not hydrological modelling. This is actually a strength of the methodology in that the true transit time distributions are permitted to be created as a consequence of any time-varying nonlinear catchment process with complete or partial mixing. However, a negative aspect is that the extreme flexibility of $K$ different nonparametric distributions is likely to give transit time distribution functions upper bounds near 1.0, unless sufficient constraints can be imposed on the form of the individual optimised distributions. There is a possibility, however, that optimising just a single nonparametric L-shaped distribution could yield useful distribution function upper bounds for time-varying situations.

25



## 1 Introduction

Kirchner (2016a,b) notes the advantages of the fraction of young water as a catchment hydrology index. Such measures have the advantage of not requiring knowledge of the form of transit time distribution upper tails, which are poorly defined by catchment tracer time series.

Kirchner (2016a) also notes the desirability of estimating transit time cumulative distribution functions as a means of identifying the form of transit time distributions. As a contribution toward this end, this technical note outlines a general nonparametric linear programming (LP) approach to obtaining upper bounds for transit time distribution functions under nonstationary conditions. The practicality of any such bounding approach is obviously related to the extent to which the

upper bounds can be located below 1.0, which in turn is dependent on an ability to provide independent constraints on transit time distribution forms. The method outlined here is therefore not a numerical recipe for calculating transit time distribution function upper bounds. Rather, it is a framework by which such bounds might be obtained with the incorporation of sufficient additional constraints.

Only the nonstationary case is considered because time variability of transit time distributions is an inevitable consequence of catchment hydrology, as noted in numerous theoretical studies and field investigations. Selected references here include Benettin et al., (2015a,b); Benettin et al., (2017); Birkel et al., (2012); Botter et al. (2010); Danesh-Yazdi et al., (2016); Harman, (2015); Heidbüchel et al., (2012); Hrachowitz et al., (2010); McMillan et al., (2012); Peralta-Tapia et al. (2016); Rigon et al., (2016);  Selle et al. (2015); Van der Velde et al., (2012); and Van der Velde et al., (2014).

## 2 Definitions

Transit time is with reference to catchment system tracer particles such as environmental isotopes, which exit a catchment as part of water discharge at some observation point at the lower end of the catchment. For any one particle departing at the observation point, its transit time is the time the particle has spent in the catchment system between entry somewhere in the catchment and exit at the observation point. There may be just some proportion $p$ of the particles entering the catchment

which actually pass out at the observation point, the rest being lost to processes such as evaporation, chemical transformation, or groundwater leakage. It is assumed here that $p$ remains constant with time. A transit time distribution is symbolized as $f_\tau(t)$ , defined to be the probability density function for transit times which applies with reference to a set of tracer particles which all enter the catchment at a given time $\tau$, and then all later exit at variable times via the observation point.






Tracer entry to the catchment is approximated as a sequence of discrete pulses $X_\tau$ of variable magnitude but equally spaced in time. For convenience the pulses are taken here as occurring at integer times. The pulse magnitudes in practice would be obtained as the product of tracer flux and tracer concentration.

A pulse of nonzero tracer magnitude at time $\tau$ is associated with a unique discrete transit time distribution $f_\tau(t)$, where the $\tau$ subscript serves to reference both the time of initiation of the transit time distribution and to denote that each distribution is different. For the purposes of the LP optimisation, an $f_\tau(t)$ distribution is expressed as a bounded discrete distribution defined on the integers over the time range $t = \tau,\ \tau+1,\ \tau+2\ \dots\ \tau+M,$ so each transit time distribution is defined over $M+1$ integer values.

Denoting $t = 0$ as a time point in the past prior to data collection, the flux-weighted tracer $Y_t$ at the observation point at time $t$ is given by the sum of the contributions by various portions of the different transit time distributions, taking into account the possible constant loss proportion $1-p$ and the previous $X_\tau$ magnitudes:

$$Y_t \;\; = \;\; p \sum_{\tau=t-M}^{t} X_\tau\, f_\tau(t) \tag{1}$$

A sequence of $K$ different transit time distributions is now considered, initiated at input pulse times $\tau_1,\ \tau_2,\ \dots\ \tau_K$. The associated $K$ transit time distributions are then envisaged as having been translated back along the time line to all have a common origin at $t = 0$. A similar translation to zero time and superposition was used by Heidbüchel et al. (2012) as a means

of working with a finite number of time-varying transit time distributions.

Using the symbolism $u(t;\ \tau_i)$ to denote the distribution $f_{\tau_i}(t)$ after translation to zero origin, the total amount of tracer particles $v(t)$ which have transit time less than or equal to time $t$ measured from zero is given by :

$$v(t) \;\; = \;\; p \sum_{i=1}^{K} X_{\tau_i} \sum_{r=0}^{t} u(r;\tau_i) \qquad 0 \le t \le M \tag{2}$$

noting that $u(0;\ \tau) = 0$.

The collective transit time distribution function $V(t)$ which incorporates all the $K$ different transit time distributions is thus given by:





$$V(t) = v(t)p^{-1} / \sum_{i=1}^{K} X_{\tau_i} \qquad\qquad 0 \le t \le M \qquad\qquad (3)$$

which is the distribution function of a weighted *K*-component finite mixture distribution where the respective *X* values are weights.

No assumptions are made concerning the hydrological mechanisms giving the *K* different transit time distributions, which might arise as a consequence of some complex time-varying nonlinear hydrological process with complete or partial mixing (Van der Velde et al., 2012).

### 3 Linear programming optimisation

The goal is to obtain nonparametric upper bound values for *V(t)* for various *t*. This is achieved for given *t* by finding the largest possible value of *V(t)*, subject to keeping an acceptable degree of data matching with the *Y* time series.

Specifically, for a given *t* the magnitude of *V(t)* is maximised by linear programming, subject to a linear goodness of fit constraint:

$$\sum_{t} \left| Y_t - Y_{t(obs)} \right| \quad \le \quad b \qquad\qquad (4)$$

where *b* is an upper bound to the sum of absolute deviations between flux-weighted observed and model tracer amounts. Alternatively, Eq. (4) can be expressed as a mean absolute deviation upper bound. The apparent product term between *p* and $u(r;\tau_i)$ is avoided by combining these quantities to produce a set of new linear variables.

For time-varying transit time distributions there is less data control at the beginning and end of the time period of the sequence of the *K* distributions, so it is helpful for all transit time distributions to be constrained in optimisation as much as physical reality permits.

The LP maximisation process is repeated for each *t*, giving a sequence of different upper bounds. The parameter *p* is
assumed unknown and is re-estimated for each *t* within the LP process to ensure the maximum extent of the upper bound.

The LP optimisation as defined above will most likely give unhelpful distribution function upper bounds somewhere near 1.0. This is because of the extreme flexibility of having *K* arbitrary nonparametric discrete distributions, enabling near-perfect matching to recorded *Y* data while still allowing 100% of tracer to be younger than a given *t*. The degree to which the
method yields upper bounds of practical value will therefore depend on the extent to which additional linear constraints can



## 4 Illustration

An uninformative cycle-free sequence of X values was simulated as 200 random variables from a normal distribution with mean 20 and standard deviation 4. For the purposes of simulating an output sequence of Y values at catchment exit, the time of input of each $X_t$ pulse marked the time of initiation of a lognormal transit time distribution. This yields 200 different lognormal distributions with different parameter values. These transit time distributions convolute the input X values to give the simulated Y data, with p in this case fixed at 1.0. This simple Y data simulation has each tracer cohort moving

independently and so is somewhat artificial. However, this is only a convenient means of generating a Y example data set. As noted earlier, no assumptions are made with respect to the origin of the true catchment transit time distributions which may be influenced, for example, by time-varying nonlinear mixing effects.

The choice of unimodal distributions to represent transit time distributions for data simulation may seem anomalous in the

light of widespread use in the literature of gamma transit time distributions with shape parameter values ≤ 1. This includes the exponential distribution as a special case, with heavier-tailed distributions being modelled as gamma distributions with shape parameters < 1 (Kirchner et al. 2000; Kirchner et al. 2001; Godsey et al., 2010; Hrachowitz et al., 2010). In reality, transit time distributions must always have at least one mode so the use of unimodal distributions for simulating data would seem appropriate, provided mode values are not too far from zero. There is no specific advantage for the use of lognormal

distributions over, say, inverse Gaussian distributions. The only desired requirement for the example was for a sharp distribution peak somewhat greater than zero and a long right tail. This actually excludes all gamma distributions.

With reference to the lognormal distribution forms, a lognormal probability density function w(x) can be expressed:

$$w(x) = [x\sigma(2\pi)^{0.5}]^{-1} \exp - [0.5(\ln x - \mu)^2 / \sigma^2] \qquad (5)$$

with parameters $\mu$ and $\sigma^2$.

The sequence of time-varying lognormal transit time distributions was created by generating $\sigma^2$ values as 200 independent random variables from a uniform distribution over the range $1 \leq \sigma^2 \leq 1.2$. The lognormal median values $m_d$ were generated as

200 independent random variables from a uniform distribution over the range $2 \leq m_d \leq 4$. In this way a simulated transit time distribution has shape and scale parameters independent of the distributions before and after in the sequence. The restricted range of variability of the simulated parameters ensured that all the distributions have long tails with modes not far removed





from zero, while still giving a degree of variability for both scale and distribution form. Fig. 1 shows the lognormal distributions from this parameter range which respectively have modes closest and furthest from zero. All the other distributions have modes intermediate between these two end point distributions.

The 200 lognormal distributions were then all converted to discrete probability distributions over a bounded range of integers with $M = 30$. That is, each distribution's probabilities are defined over a finite range of integer values $t$, where $0 \leq t \leq 30$. The value of $M = 30$ here is arbitrary but $M$ need only be sufficiently large so as not to affect the outcome of the LP optimisation. This will be evident following the optimisation if the upper tail values of the optimised transit time distribution are all zero. The value of $M$ may become more critical if seeking distribution function lower bounds (Section 5).

Fig. 2 shows the 200 values of the simulated input $X$ time series as well as the output $Y$ time series obtained from the convolution of the $X$ values with the discrete transit time distributions. Of the 200 discrete distributions, the time sequence of 100 distributions from distribution 51 to distribution 150 was selected as the basis for estimating the upper distribution function bounds. The $Y$ values created just from the associated $X_{51}$ to $X_{150}$ input pulses are plotted separately in Fig. 2.

Application of Eq. (3) gives the true 100-distribution combined cumulative distribution function as plotted in Fig. 3, for which an upper bound is sought. In this case the cumulative distribution function with $X$ weightings is for practical purposes indistinguishable from the unweighted equivalent.

Prior to LP maximisation of a given $V(t)$, additional linear constraints were set up such that the all the optimised transit time distributions must have the property

$$f_\tau(t) \geq f_\tau(t+1), \qquad f_\tau(t) - f_\tau(t+1) \geq f_\tau(t+1) - f_\tau(t+2) \qquad (6)$$

which is the discrete equivalent of a continuous probability density function having a negative first derivative and positive second derivative. This forced L-shaped discrete distribution form for $t \geq 1$ has the advantage of avoiding the requirement to seek distribution mode values as part of the optimisation.

Taking all the various constraints and variables into account, the defined LP optimisation problem for given $t$ involved
solving for about 5,000 linear variables subject to around 10,000 linear constraints. This is not a large optimisation by current LP standards and solving to obtain an upper bound for given $t$ took only a few seconds on a standard PC using a commercial Excel LP utility.



It happened in this example that the sequence of simulated $X$ data as normal random variables was in fact insufficient to provide useful upper bounds. This is because there is still sufficient flexibility in the 100 constrained L-shaped nonparametric distributions such that the distribution function upper bounds are never far from 1.0 (Fig. 3), even when perfect matching to the $Y$ data was a specified requirement. This flexibility issue may or may not arise generally for less

random $X$ data, such as incorporating a seasonal cycle.

As an alternative, a more significant imposed level of constraint is to force all 100 nonparametric optimised distributions to be a single L-shaped discrete distribution which does not change over time. This gives a more encouraging result with the upper bound now being close to the true distribution function up to a distribution function value of about 0.6 (Fig. 3). This is

despite the time-variability of the actual transit time distributions which were used to simulate the $Y$ data. In this case the goodness of fit to the $Y$ data was constrained to have a mean absolute deviation no greater than 1.3, which maintains reasonable fits to the simulated data although individual peaks are not accurately matched. Fig. 4a shows model and data comparison from the 1.3 constraint after maximising the portion of young tracer of age $t \leq 2$.

The reduced fitting flexibility when going from 100 constrained nonparametric distributions to just a single constrained nonparametric distribution is reflected in the impossibility of forcing a perfect fit to the data for any $t$. Attempting a fit constraint with mean absolute deviation < 1.3 created an infeasible LP maximisation in this example.

The transit time distribution function upper bound obtained by optimising a single nonparametric discrete distribution was

robust against a degree of measurement error, simulated in this case by taking each $X$ and $Y$ data point and adding to them an independent random variable from a uniform distribution over the range -2 to 2. The degree of achievable fit to the modified $Y$ data was not as good as before (Fig. 4b), as would be expected. In this case an infeasible LP maximisation arose when attempting to constrain the mean absolute deviation to any value less than 2.0. However, despite the induced noise and reduced fit, the resulting upper bounds are not far removed from those obtained from the noise-free situation (Fig. 3).


The optimised L-shaped parametric distributions for various $t$ in the single-distribution case have somewhat unrealistic geometric forms (Fig. 5), arising from the nature of the constraints imposed by Eq. (6). However, the upper tail values were all zero, indicating the choice of $M = 30$ was sufficiently large not influence the result. Interestingly, comparison of Fig. 1 and Fig. 5 shows that an approximation to $Y$ data can be maintained without the optimised transit time distribution having a

long right tail, even though all the different transit time distributions in fact had long tails.

The results from the single nonparametric distribution raised the possibility that perhaps upper bounds to transit time distribution functions for time-varying situations could be achieved by optimising just one nonparametric distribution. However, there is no suggestion that a strong argument has been made here. This is because the utilised linear mixture-free





process of simulating the *Y* data would be expected to be amenable to approximation by a single nonparametric distribution which is roughly representative of the 100 time-varying transit time distributions. An obvious next step in the evaluation of the method would be to test the single-distribution approach against a range of simulated time-varying tracer data sets obtained from different nonlinear models. A starting point might be the simple two-store model described by Kirchner

(2016b).

## 5 Discussion and conclusion

Parametric transit time distributions have been widely utilised, usually arising in the context of assumed hydrological models. Overviews are given by Leray et al. (2016), and McGuire and McDonnell (2006). Nonparametric distributions are

less utilised but do appear from time to time in the literature. Previous work mentioning various aspects of nonparametric transit time distributions includes Cirpka et al. (2007), Liao et al. (2014), McCallum et al. (2014), and Visser et al. (2013). The present paper would appear to be the first to propose nonparametric distributions in combination with linear programming as a mechanism which might yield bounds to time-varying transit time distribution functions.

Whether the linear programming approach will prove of value remains an open question and it is hoped further investigation will be encouraged. The need for confirmation with nonlinear data simulations has already been mentioned. There is also a need to test the method against recorded data sets, particularly where cyclic variation in the *X* data might aid bound definition.

It may be that the single nonparametric distribution approximation is sufficient in a number of situations. Alternatively, further constraints might be added to sets of multiple time-varying nonparametric distributions sufficient to move the upper bound closer to the true transit time distribution function. One option could be to allow time variation of just the younger part of the different nonparametric transit time distributions, but constraining the distribution tails to be similar. This was done in a parametric context by Heidbüchel et al. (2012), where gamma distribution tail behaviour was assumed for time-

varying transit time distributions. Another option might be to assume the transit time distribution parameters change over time as a Markov process.

Still another means to achieve tighter constraints around transit time distribution functions would be to use multiple tracers, with the data fitting requirement applied to multiple *Y* time series simultaneously. In this regard, a catchment study by

Kirschner et al. (2010) suggests that oxygen-18 and chloride transit time distributions may have a common shape. Suitable linear constraints might then be added to force a common form to give an improved transit time definition of both tracers.

Focus in this paper has been on upper bounds but there is also the possibility of using LP minimisation in a similar way to obtain lower bounds for time-varying transit time distribution functions. Distribution function lower bounds were briefly investigated but it became evident that for the $M = 30$ example the lower bounds were an artefact of $M$, with the optimised nonparametric distributions being simple triangular distributions bounded below at $t = 30$. However, for larger values of $M$

this effect may become negligible and useful lower bounds might be obtained for some data sets.

As with any study of catchment time-varying situations, the question arises as to the extent to which results are time-specific or can be taken as a catchment characteristic. The views of Heidbüchel et al. (2012) are of relevance here, with those authors suggesting that a finite mixture distribution of transit time distributions might be regarded as time-invariant. This is

subject to a sufficient number of incorporated distributions and an absence of climatic or catchment change.

To conclude, the nonparametric linear programming approach is offered here in the spirit of drawing attention to an unfinished entity which might prove a useful means of moving toward solving the important hydrological problem of transit time distribution definition in the context of time-variability. However, considerable further investigation is still required if

the practicality of the method is to be verified.

## Acknowledgement

This paper has been developed in the light of a range of helpful comments by four anonymous reviewers on a previous

submission.

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



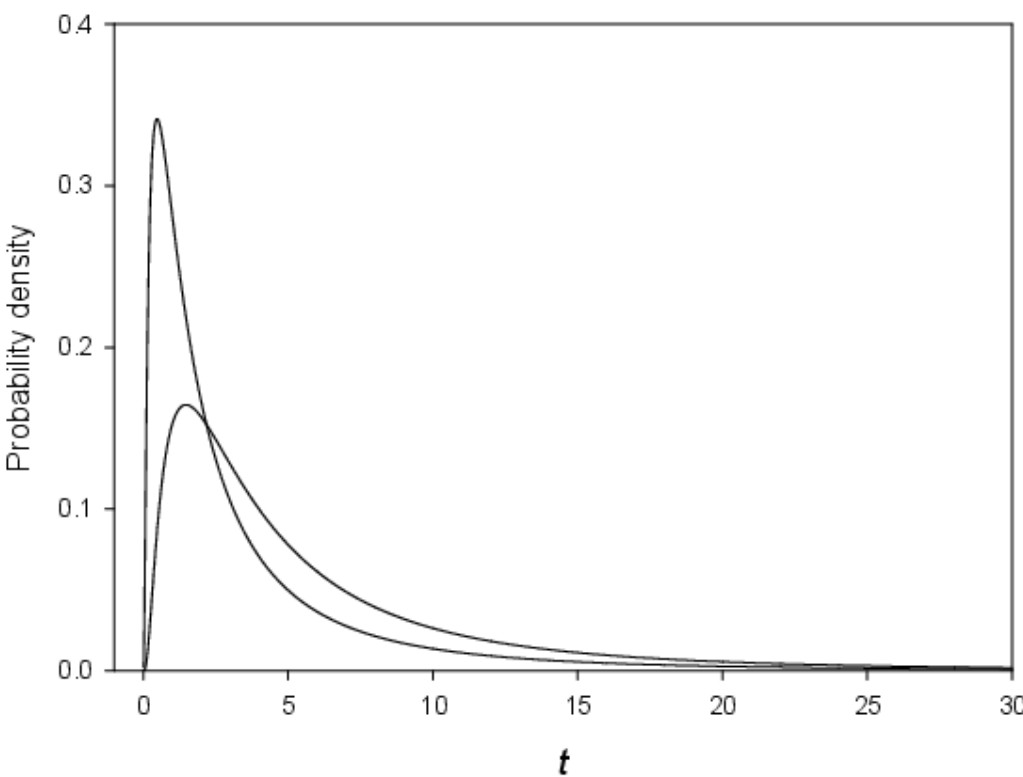

**Figure 1: The two lognormal transit time distributions giving respective modes nearest and furthest from zero.**



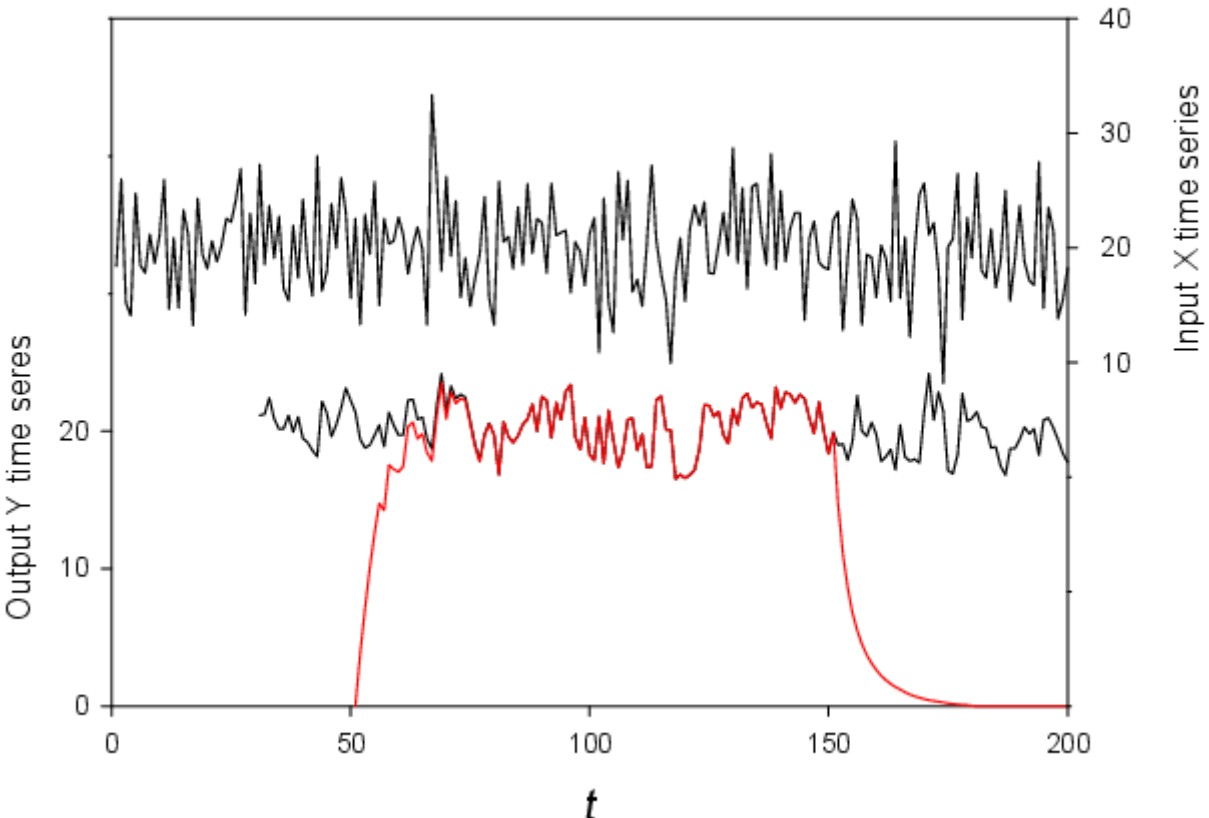

**Figure 2: Time series plots. The upper plot is the 200 simulated input *X* values of tracer. The lower plot (black) is the simulated output *Y* values as obtained by convoluting the *X* values with the lognormal transit time distributions. The warm-up portion of the time series is not plotted. The red plot shows the contribution to the *Y* data from the *X* input pulses 51-150. The *X* and *Y* simulated data are plotted on different axes for ease of viewing.**





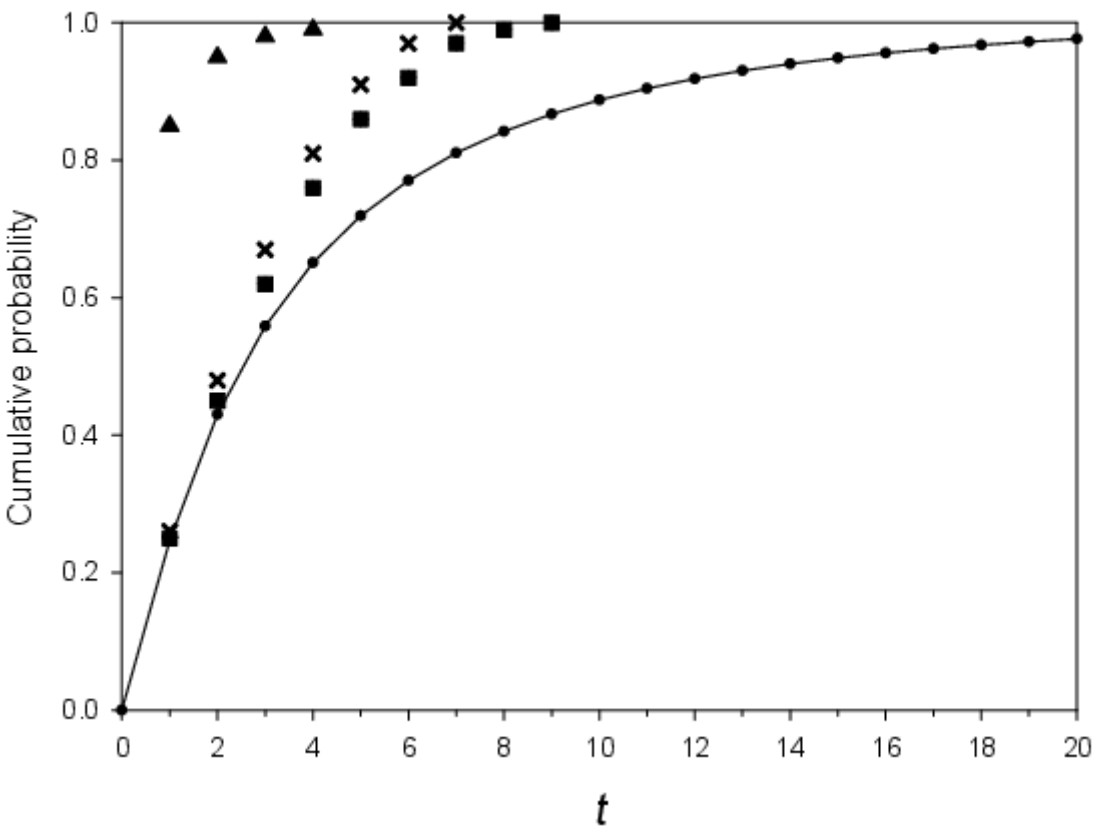

**Figure 3: Transit time cumulative distribution function and upper bounds. The rising line shows the collective cumulative distribution function of the time-varying nonparametric discrete distributions for transit time distributions 51-150. Triangles are distribution function upper bounds as obtained from 100 different simultaneously optimised transit time distributions. Crosses are upper bounds as obtained from a single optimised nonparametric transit time distribution. Squares are upper bounds as obtained from a single optimised nonparametric transit time distribution, where random noise has been added to the $X$ and $Y$ time series. All optimised distributions were constrained to be L-shaped for $t \geq 1$.**


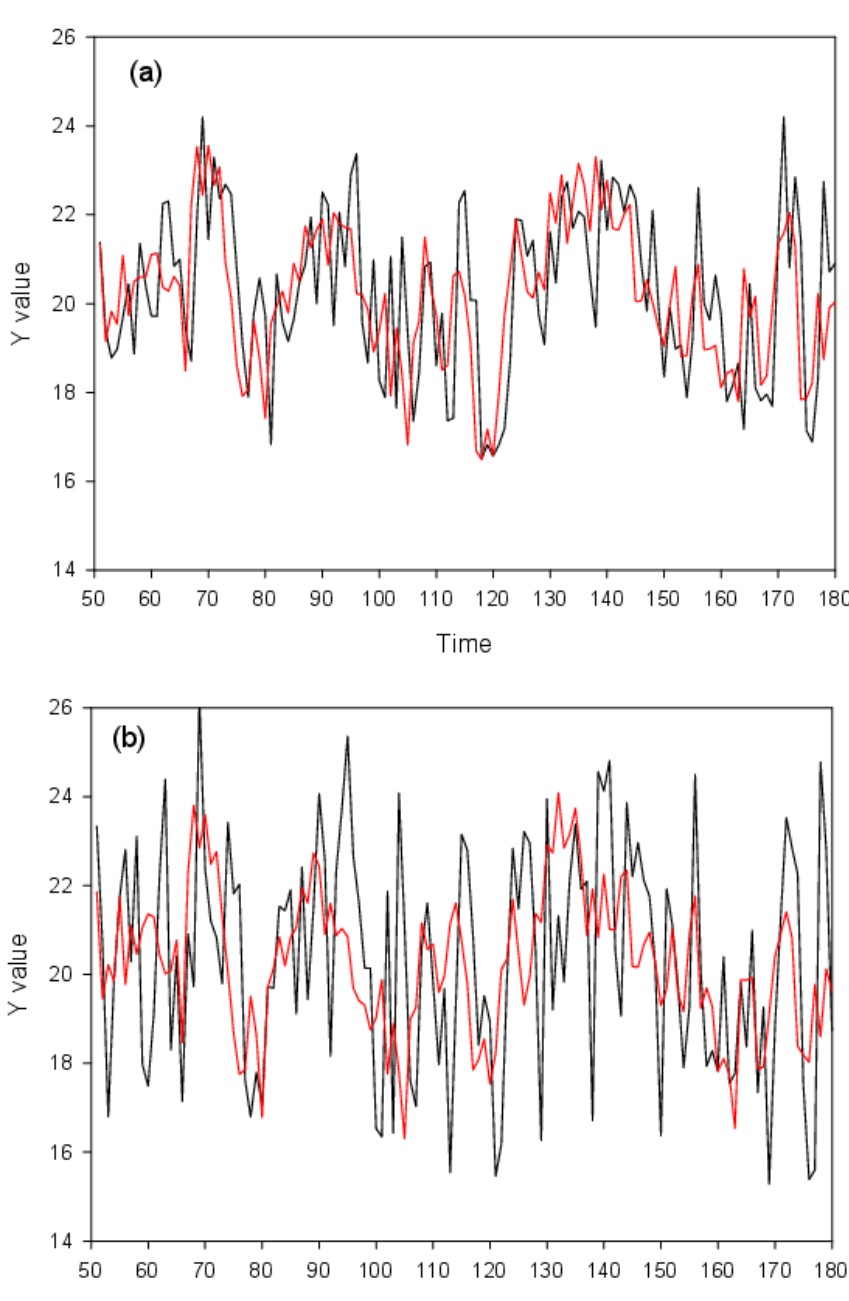

**Figure 4: Model values (red) and simulated *Y* data time series (black) for the particular case of maximising the proportion of tracer age ≤ 2, with a single nonparametric L-shaped distribution. (a) shows the case without simulated measurement error (mean absolute deviation constrained to be ≤ 1.3), and (b) shows the case with random error added to both the *X* and *Y* simulated data (mean absolute deviation constrained to be ≤ 2).**



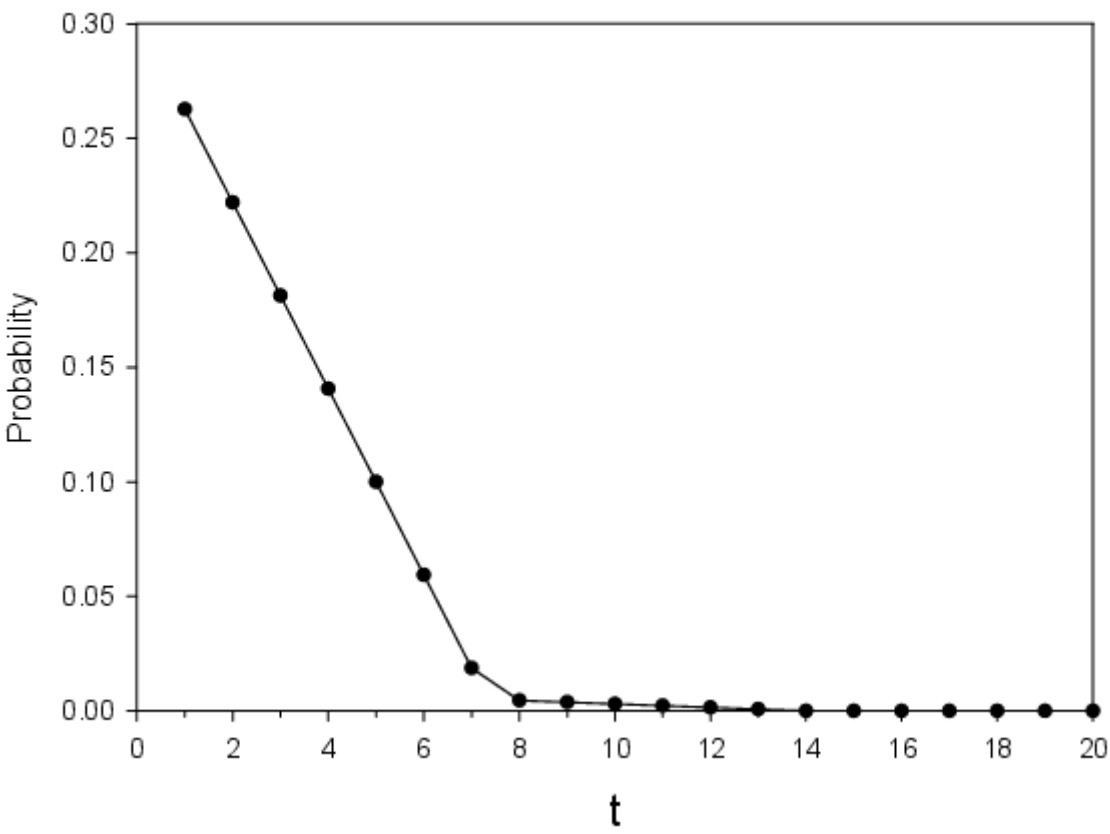

**Figure 5: Optimised discrete transit time distribution for the case described in Fig. 4(a), obtained from maximizing the proportion of tracer age ≤ 2. The sum of the first two probability values (0.26 and 0.22) gives the 0.48 value corresponding to the cross symbol for *t* = 2 in Fig. 3.**

