# Peer review of "A nonparametric approach toward upper bounds to transit time distribution functions"

_Hydrology and Earth System Sciences, 2017_

## Referee Comment (RC1) · Anonymous Referee #1 · 17 Aug 2017

The paper proposes using a simple nonparametric linear programing approach as an optional method to obtain upper bounds of transit time. The study was done using a random input dataset of 200 unit steps of time. The subject of transit time of great relevance in hydrology, and searching for newer methods can bring rewards to the field. The paper was well written, but there are some points I would like to address on this review that concern me on the usefulness of the approach. General comments: 1. Even when there is a section called "discussion and conclusion" I found there was some discussion found on the earlier section Illustration, especially by the end of this one. 2. Assuming a constant value for the lost water to evaporation and others can lead to a biased result, especially in catchments with strongly defined seasons were evaporation is far from constant. 3. It was used a time variant dataset, however,

the aim of the results were time invariant as I understood. For 100 "days" there was only one cumulative probability curve and three different approaches of upper bound calculations. Knowing that many systems have a large span of transit time distributions throughout a year, what would be the contribution of knowing the upper bound of the whole year? Knowing that this bound could be pushed by only one extreme day and not be representative of the rest of the year. 4. Following the previous comment, could you calculate the upper bound continuously to each time step separately using LP? Specific comments: 1. Page 2 Line 9: Define what you mean with "1.0". 2. Page 3 Eq 1: So $K = M + 1$? If so, it should be said on the text to make the read easier. 3. Page 7: While discussing the results on Figure 4 there is no metric on how the figures changed rather than only visual. How can the reader measure the goodness? How was the fit changed from Fig 4a to Fig 4b? 4. Page 8 Line 16: "...has already been mentioned." When? Reference? Or earlier on the text? Technical corrections: 1. Page 6 Line 20: Delete the extra "the" on the sentence "...were set up such that the all the optimised..."

---

## Referee Comment (RC2) · Anonymous Referee #2 · 17 Aug 2017

The author develops a nonparametric method to determine upper bounds in catchment transit time distributions. Upper bounds are found through an optimization technique whose objective is to reproduce the output to a certain level of accuracy. I think that the article can be of interest to the readership of this journal, but a few points still need to be clarified to make it more accessible to the community. I therefore suggest "major revisions".

First of all, the method is very general and lacks of physical interpretation. I understand this is a technical note so I only suggest the authors to use a real hydrologic example as illustration (Section 4). My opinion is that the current illustration does not introduce anything new and an application to real tracer data would be more useful.

Second of all, assuming that "p" remains constant and neglecting the transit time dis-

tributions of other forms of output (e.g., evapotranspiration) is problematic in real applications (see for example the papers by Botter, 2011, Harman, 2015 and by Calabrese, 2017). Can the method introduced in this manuscript be extended to introduce multiple outputs, each one with its own transit time distribution?

Lastly, how is this method better than the estimation of age thresholds discussed in Kirchner (2016a)? I ask the author to discuss this.

Minor comments:

-Page 3, line 1. $X_\tau$ should be placed after "magnitude".

-Page 3, line 3. Shouldn't "tracer flux" be "water flux"?

-Page 3. The paragraph which begins with "A sequence of K different.." needs to be expanded. It is hard to follow the translation in time and the new notation. Why not introducing the notation with common origin at t=0 from the beginning?

-I think Figure 1 is not necessary.

References: -Botter, G., E. Bertuzzo, and A. Rinaldo (2011), Catchment residence and travel time distributions: The master equation, Geophys. Res. Lett., 38, L11403, doi:10.1029/2011GL047666. -Harman, C. J. (2015), Time-variable transit time distributions and transport: Theory and application to storage-dependent transport of chloride in a watershed, Water Resour. Res., 51, 1–30, doi:10.1002/2014WR015707. -Calabrese, S., and A. Porporato (2017), Multiple outflows, spatial components, and nonlinearities in age theory, Water Resour. Res., 53, 110–126, doi:10.1002/2016WR019227.

---

## Author Comment (AC1) · 25 Aug 2017

My thanks to both referees for their comments.

Some specific comments from the referees are responded to below.

**Referee #1**

The aim is not to obtain a time-invariant result. The distribution function upper bound is with respect to a finite mixture of transit time distributions over some specific time period, which may or may not be representative of other time periods.

There is no suggestion of seeking to define an upper bound to the distribution function of a single transit time distribution among all others – probably an impossible task outside of a large tracer experiment.

In the simulation it happened that the simulated data using a time sequence of 100 different distributions could be approximated by a single transit distribution. This may or may not be a general result. As noted in the paper, it might simply be an artefact of the data simulation method.

The approach is to seek an upper bound to the distribution function of a finite mixture distribution comprised of some number of true transit time distributions. This finite mixture distribution can be thought of as an average of a sequence of transit time distributions, so will not be dominated by the effect of any one distribution. This averaging of course does obscure any distribution-to-distribution differences, but the upper bound still has potential to give some information specific to that catchment which might be compared to other catchments.

**Referee #2**

Almost by definition, nonparametric approaches have limited capability of physical interpretation. That is, nonparametric distributions here have ability to approximate any time sequence of transit time distribution forms, but can offer no explanation as to origin of forms. This is both a strength and weakness, as noted in the paper.

At this stage, my feeling is that it would not be helpful to use real data. The reason is that the approach is still only a suggestion for possible development. With real data we never know what the true transit time distributions are. The advantage of simulated data is we can measure against truth. There is in fact a need for considerably more simulations for the approach to be demonstrated as potentially useful. Hopefully this short paper as a technical note may encourage such further numerical work.

The method could be extended to multiple outputs, provided there are tracer time series of those multiple outputs to serve as reference.

**Effect of time-varying p**

Both reviewers raised the desirability of being able to allow for time-varying p. Effectively this means allowing for varying weights in the finite mixture distribution. There is no disagreement that this would better approximate reality, but would require reworking of the LP setup.

I this regard, I think referee #2 may have been a little too kind in requesting a major revision. I would like therefore to withdraw the paper at this point, and (if the Editor permits) submit a new paper at a later date, incorporating the effect of variable p.

Both referees also raised a number of other points. These are not explicitly responded to here but note would be taken of those various comments in working up any new paper for review.

---

## Referee Comment (RC3) · M. Hrachowitz (Referee) · 4 Sep 2017

The manuscript "A nonparametric approach toward upper bounds to transit time distribution functions" by Beardsley addresses the problem of these upper bounds being difficult to constrain with the typically available data and which may result in the inference of transit time distributions that can considerably misrepresent the real system characteristics.

The overall topic of this manuscript is of considerable interest as a technique allowing to efficiently constrain the upper bounds of transit time distributions would be extremely valuable to develop a better understanding of the underlying processes. In spite of the general interest, the manuscript remains somewhat superficial and the presented anal-

ysis and results are not entirely convincing. It thus may be beneficial to go a bit more into depth and develop the manuscript a bit further to the point, for example by adding some more case studies. This could be done either (1) extending the existing analysis by further toy models, to provide insights how different assumptions and boundary conditions affect the method and/or (2) ideally provide a demonstration with real data.

Specific comments:

(1) P.2, l.1-19: A problem statement that is a bit more detailed would help the reader to better understand the relevance of the analysis in this manuscript (in other words, try to place more emphasis on the sentence in line 12-13). Likewise, it would be good to formulate an explicit science question here and provide a working hypothesis that is going to be tested in the manuscript.

(2) P.2, l.6: why such an emphasis on "cumulative"? The CDF should be known if the PDF is known and vice versa. Please clarify.

(3) P.2,l.8: Please be a bit more specific to avoid misunderstandings. What is exactly meant by "upper bounds"? Feasible and physically meaningful bounds to the tails of these distributions?

(4) P.2,l.9-10: not entirely clear what is meant by "...the extent to which the upper bounds can be located below 1.0,...". Please rephrase.

(5) P.2,l.18: Hrachowitz et al. (2013; HESS) and/or (2015;Hydrological Processes) would fit better here than the 2010 reference

(6) P.2,l.21-22: I am not sure, if this statement (and the emphasis on tracers thereafter) is sufficiently exact. In my understanding, it does mix up general concepts with real world applications. Tracers are essentially tools. Thus in the first instance, transit times are with reference to the movement of individual *water molecules* (which, in reality, and with the available observation technology can only be tracked with tracers).

(7) P.2,l.26: it needs at least to be acknowledged that the assumption of p being constant does not hold for real world systems (as demonstrated by e.g. Harman, 2015; WRR; Hrachowitz et al., 2015; Hydrological Processes).

(8) P.2,l.26-29: see comment (6) – the emphasis needs to be the movement of water molecules, which are tracked with the help of tracers, assuming these tracers are conservative and move with the water.

(9) P.3,l.1: see (6) and (8)

(10) P.3,l.3: pulse magnitude=flux*concentration? i.e. in SI unit symbols $(L^3*M/L^3)*(M/L^3)=M^2/L^3$. This does not make sense. Please correct this typo. I suppose what is meant is pulse magnitude=tracer mass flux=water volume*tracer concentration, i.e. $M=L^3*M/L^3$.

(11) P.3,l.13: no, a constant loss proportion is effectively NOT possible, given the natural variability in environmental systems.

(12) P.4,l.22: How is it evaluated/decided if it is permitted by reality? What is meant by "reality" here?

(13) P.4,l.24: I cannot fully follow here. I thought p is kept constant.

(14) P.5,l.5-30: It would be good to clarify/discuss in how far this approach is different to the different approaches suggested by Heidbuechel et al. (2012; WRR) and Hrachowitz et al. (2010; WRR) – both already in the references.

(15) P.5,l.14-15: This statement is confusing. Distributions from the exponential family (also including gamma with shape parameter <1) are also unimodal. Strictly spoken, the mode of a continuous probability distribution is the value at which the probability density function has its maximum value (which is clearly defined for exponential family distributions). Please rephrase.

(16) P.5,l.18: firstly, see (15) – thus, the mentioned gamma distributions *are* unimodal. What is obviously meant here is modes that are found at x>0+epsilon. And

secondly, in addition, the statement also depends on the (time) scale and (time) interval of interest. While a distribution with a mode at x>0+epsilon is likely needed for a high temporal resolution (e.g. <15minutes or so), most models so far were, as dictated by the available data, implemented at temporal resolutions much higher than that. For such higher resolutions, the delayed mode can typically not be resolved anymore by the available data, thus resulting in the necessity of using exponential- or gamma distributions (shape factor<1). Please rephrase.

(17) P.6,l.13: why 51 to 150? Please clarify.

(18) P.7,l.26: again – what is meant by unrealistic? On basis of what is this judged and what is actually meant by "realistic"?

Best regards, Markus Hrachowitz

---

## Author Comment (AC2) · 6 Sep 2017

My thanks Markus, for your detailed and helpful comments.

As noted in my previous response, my intention is to withdraw the manuscript and (if the Editor permits) submit a new paper at a later date explicitly incorporating the effect of varying p, while also taking into account all current reviewer comments. My feeling was that the paper would then have such a different look to it that it would be better considered as a new submission. I hope the present reviewers will remain, if they are kind enough to do that.

As you note, there is a need for making a better case. Nonparametric methods are very flexible and there is a need to demonstrate that matching closely to data can in fact lead to helpful upper bounds located sufficiently below 1.0. However, I would prefer not to extend beyond a Technical Note in scope. If permitted, the way to do it might be to have an electronic supplement going into more examples. My preference is toward the toy model approach in the first instance, because application to real data, doing it properly, could require adding some constraints from local knowledge. Also with toy models we know what the truth is. It might also be helpful in the supplement to include some example LP setups so others can quickly run their own data.

In your comments, I think there are only a few minor points that need a brief response or comment at this stage (below).

With respect to (2)
It is true that if a cumulative distribution is known then so too is the density function. However, I am only obtaining an upper bound to the cumulative distribution and the derivative of an upper bound expression cannot be interpreted as providing information about the density function.

With respect to (4)
Cumulative distribution functions are upper-bounded at 1.0 by definition. So if, for given t, an upper bound of, say, 0.99 was obtained, then this would not be regarded as useful.

With respect to (15&16)
Gamma distributions only have defined modes for $\alpha \geq 1$. See, for example, the chapter on the gamma distribution in Johnson et al (1994). But yes – I should have mentioned explicitly that I am concerned with modes > 0 because for the special case of the exponential distribution there is a mode at zero.
Simulations should be based on the correct form of transit time distributions f(t) such that f(0)=0, independent of considerations of instrumental resolution. So there is no necessity to use L-shaped distribution forms – just the opposite in fact. See also doi.org/10.5194/hess-2017-497 and discussion.

My thanks again for your detailed input.

Johnson, N.L., Kotz, S., Balakrishnan, N. (1994) Continuous Univariate Distributions V1 (2nd Ed)

---

## Referee Comment (RC4) · Anonymous Referee #4 · 19 Sep 2017

This is a resubmission of a previusly submitted manuscritpt on the same topic. The paper has been modified to include nonstationarity and expand the reference section. With these respects, the paper certainly does a better job than the previously submitted paper. However, I'm still a bit puzzled by this paper and his relevance for the hydrologic community, for the following reasons.

HESS is an hydrological journal, which is mainly read by hydrologists. Therefore, I think the author should do much more efforts to better convey his message to the hydrological community. The language, the way of presenting, the discussion of the implications of the findings are a bit technical and maybe somewhat at odds with the literature on the topic. Honestly, it's really difficult to grasp the relevance of this work for the hydrological community given the example presented in the paper. The author is firm in suggesting

that there is no hydrological modeling in the paper, which is good. On my side, as an hydrologist, I have to say that I had really hard time in understanding what is going on in this paper. Using dimensionelss quantities instead of physically-meaningful variables (concentrations, times, etc) and working only with synthetic examples that have nothing to do with real world hydrological systems does not help. I think the only way to be more convincing about the usefulness of his approach is to apply the framework to real world concentration data (there are freely available datasets on the web), and provide upper bounds of travel time distributions in contexts where you have to face data gaps and limited timeseries, and in cases previous estimates of mean travel times and travel time distributions are available. This is not to state that there is no value in the procedure. However, I think the application of the theory to reality would definitely offer the opportunity to explain the method and its potential much better, and show its advantages with respect to other low-complexity parametric approaches allowing the estimate of travel time distributions based on hydrochemical data.

---

## Author Comment (AC3) · 19 Sep 2017

My thanks to Referee 4 for the comments.

If the paper is not clear to hydrologists then that is certainly undesirable from my viewpoint.

Some of the points raised are similar to the other reviewers so I have little further comment to make beyond what has been said before, and I am not in any substantial disagreement.

The desirability of a real-world data set is noted. However, it is important to keep a synthetic data set also because only in this way is there a "truth" to measure the method against, artificial though it may be.